# Light-driven C–H activation mediated by 2D transition metal dichalcogenides

Jingang Li [1,2], Di Zhang[3], Zhongyuan Guo [3], Zhihan Chen[1], Xi Jiang[4], Jonathan M. Larson [5], Haoyue Zhu[6], Tianyi Zhang[6], Yuqian Gu[7], Brian W. Blankenship [2], Min Chen[8], Zilong Wu[1], Suichu Huang[1], Robert Kostecki[9], Andrew M. Minor [8,10], Costas P. Grigoropoulos [2], Deji Akinwande[7], Mauricio Terrones[6,11,12], Joan M. Redwing [6,13], Hao Li [3] ✉ & Yuebing Zheng [1] ✉

C–H bond activation enables the facile synthesis of new chemicals. While C–H activation in short-chain alkanes has been widely investigated, it remains largely unexplored for long-chain organic molecules. Here, we report light-driven C–H activation in complex organic materials mediated by 2D transition metal dichalcogenides (TMDCs) and the resultant solid-state synthesis of luminescent carbon dots in a spatially-resolved fashion. We unravel the efficient H adsorption and a lowered energy barrier of C–C coupling mediated by 2D TMDCs to promote C–H activation and carbon dots synthesis. Our results shed light on 2D materials for C–H activation in organic compounds for applications in organic chemistry, environmental remediation, and photonic materials.

The emergence of C–H bond activation has provided revolutionary opportunities in organic chemistry, materials science, and biomedical engineering[1]. Specifically, the activation and functionalization of the ubiquitous C–H bonds enable new synthetic routes for functional molecules in a more straightforward and atom-economical way[2–5]. Since C–H bonds are thermodynamically strong and kinetically inert[6], many catalysts have been developed for C–H activation, including transition metals (e.g., palladium[7], cobalt[8], and gold[9,10]), zeolites[11,12], and metal-organic frameworks[13,14].

While intensive research efforts have been focused on C–H bonds in short-chain alkanes (e.g., methane and ethane)[15,16] and aromatic compounds[17], C–H activation in long-chain organic

molecules is rarely reported. Yet, the derivation of C–H bonds in these complex molecules has significant potential in synthesizing functional organic complexes and transforming environmental pollutants (e.g., fossil-resource-derived hydrocarbons) into more valuable chemicals[18,19].

Herein, we report the light-driven C–H activation in long-chain molecules mediated by two-dimensional (2D) transition metal dichalcogenides (TMDCs). This TMDC-mediated C–H activation in organic molecules enables optical synthesis and patterning of luminescent carbon dots (CDs) on solid substrates. As a first example, we achieve the light-driven transformation of cetyltrimethylammonium chloride (CTAC, $C_{19}H_{42}ClN$), a long-chain quaternary ammonium surfactant[20],

[1]Materials Science & Engineering Program, Texas Materials Institute, Walker Department of Mechanical Engineering, The University of Texas at Austin, Austin, TX, USA. [2]Laser Thermal Laboratory, Department of Mechanical Engineering, University of California, Berkeley, CA, USA. [3]Advanced Institute for Materials Research (WPI-AIMR), Tohoku University, Sendai, Japan. [4]Materials Sciences Division, Lawrence Berkeley National Laboratory, Berkeley, CA, USA. [5]Department of Chemistry and Biochemistry, Baylor University, Waco, TX, USA. [6]Department of Materials Science and Engineering, The Pennsylvania State University, University Park, PA, USA. [7]Chandra Family Department of Electrical & Computer Engineering, The University of Texas at Austin, Austin, TX, USA. [8]Department of Materials Science and Engineering, University of California, Berkeley, CA, USA. [9]Energy Storage and Distributed Resources Division, Lawrence Berkeley National Laboratory, Berkeley, CA, USA. [10]National Center for Electron Microscopy, Molecular Foundry, Lawrence Berkeley National Laboratory, Berkeley, CA, USA. [11]Center for Two-Dimensional and Layered Materials, The Pennsylvania State University, University Park, PA, USA. [12]Department of Physics, Department of Chemistry, The Pennsylvania State University, University Park, PA, USA. [13]2D Crystal Consortium, Materials Research Institute, The Pennsylvania State University, University Park, PA, USA. ✉e-mail: li.hao.b8@tohoku.ac.jp; zheng@austin.utexas.edu

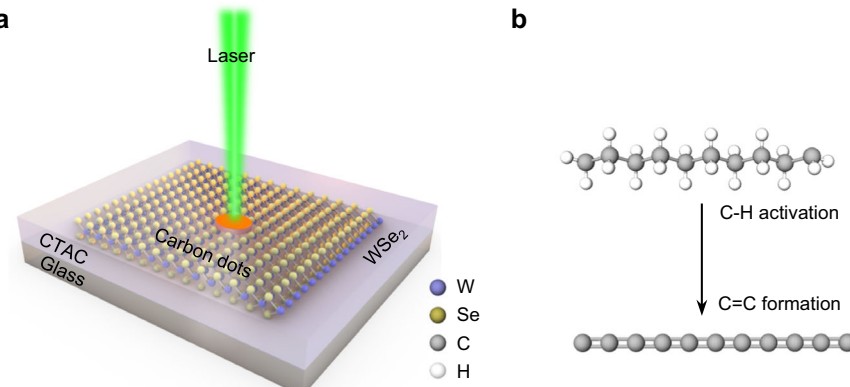

**Fig. 1 | General concept of light-driven C–H activation in long-chain molecules mediated by 2D materials. a** Schematic showing the light-driven transformation of CTAC on an atomic layer of $WSe_2$ into luminescent CDs. **b** Schematic showing the photochemical reaction process involving the activation of C–H bonds and the formation of C=C bonds.

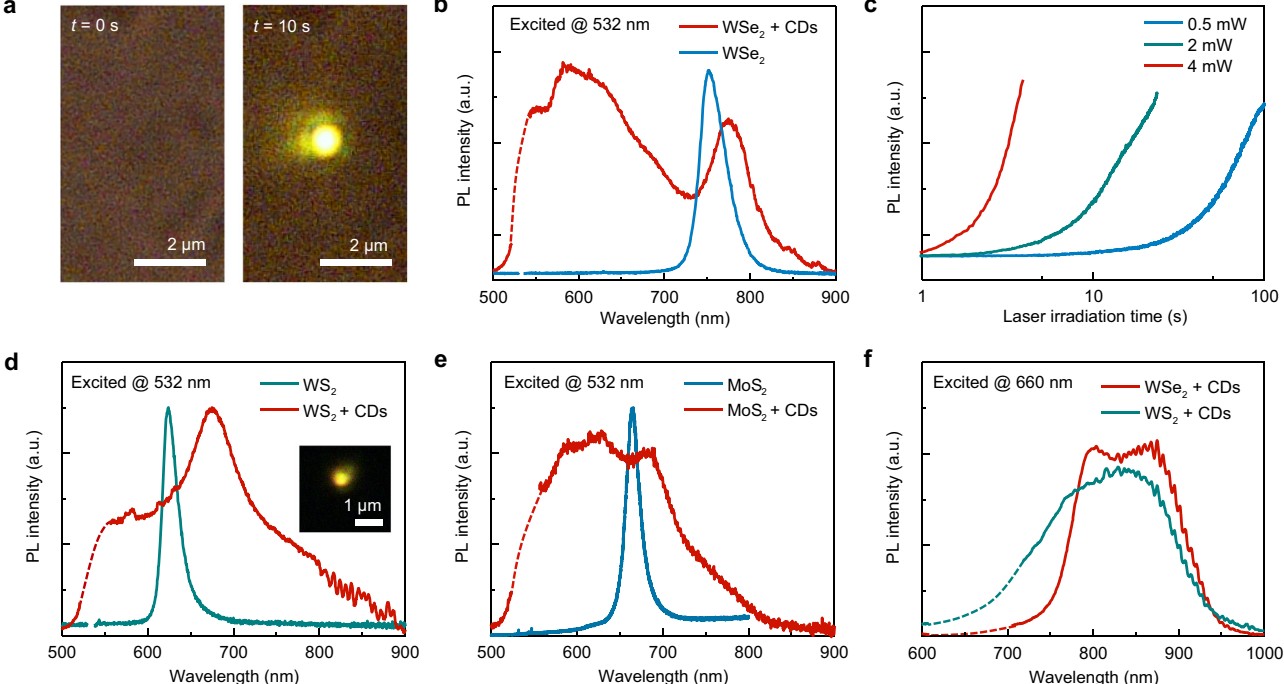

**Fig. 2 | Optical characterizations of 2D-mediated C–H activation and CD synthesis. a** Optical images showing the CTAC on the $WSe_2$ sample under a 532 nm laser irradiation at $t = 0$ s and $t = 10$ s. The laser power is 2.5 mW. The yellowish PL emission comes from the optically synthesized CDs. **b** The PL spectra of $WSe_2$ and $WSe_2$ + CDs hybrids. **c** Time-resolved PL intensity of CDs at 600 nm from the CTAC on $WSe_2$ sample under a 532 nm laser irradiation with different optical power. **d**, **e** The PL spectra of **d** $WS_2$ and $WS_2$ + CDs hybrids and **e** $MoS_2$ and $MoS_2$ + CDs hybrids under the excitation of a 532 nm laser. Inset in (**d**) optical image showing the PL emission from the $WS_2$ + CDs sample. **f** The PL spectra of $WSe_2/WS_2$ + CDs samples excited by a 660 nm laser. "a.u." in (**b**–**f**) stands for arbitrary units.

into luminescent CDs on $WSe_2$ monolayers. By coupling experiments with density functional theory (DFT) calculations, we unravel the role of Se vacancies and oxidized states of $WSe_2$ in promoting H adsorption. We further show that 2D TMDCs can facilitate the C–C coupling with a lowered energy barrier to catalyze C–H activation in complex organic molecules. This type of light-driven reaction mediated by 2D materials can be generalized to other long-chain organic compounds for the broader impacts on organic synthesis, chemical degradation, and photonics.

## Results

A typical experimental configuration is presented in Fig. 1a. A thin layer of solid CTAC is coated on a monolayer $WSe_2$ grown by chemical vapor deposition (CVD). The monolayer feature of $WSe_2$ is confirmed by the strong photoluminescence (PL) peak at ~750 nm (Fig. 2b, blue curve). Under the irradiation of a low-power continuous-wave laser (~0.2–5 mW), CTAC molecules undergo $WSe_2$-mediated C–H bond activation and the subsequent C=C bond formation (Fig. 1b). CTAC contains long carbon chains and quaternary ammonium cations, which has been commonly used as surfactants for chemical synthesis and fabric softeners[21]. Here, we choose CTAC as a first example due to its clean carbon-chain structure, solid form under ambient conditions, and wide existence in nanomaterials systems. This light-driven reaction can also be applied to other organic compounds.

The laser irradiation on hybrid CTAC/$WSe_2$ thin films leads to the emergence of bright luminescence from CDs (Fig. 2a). The evidence of

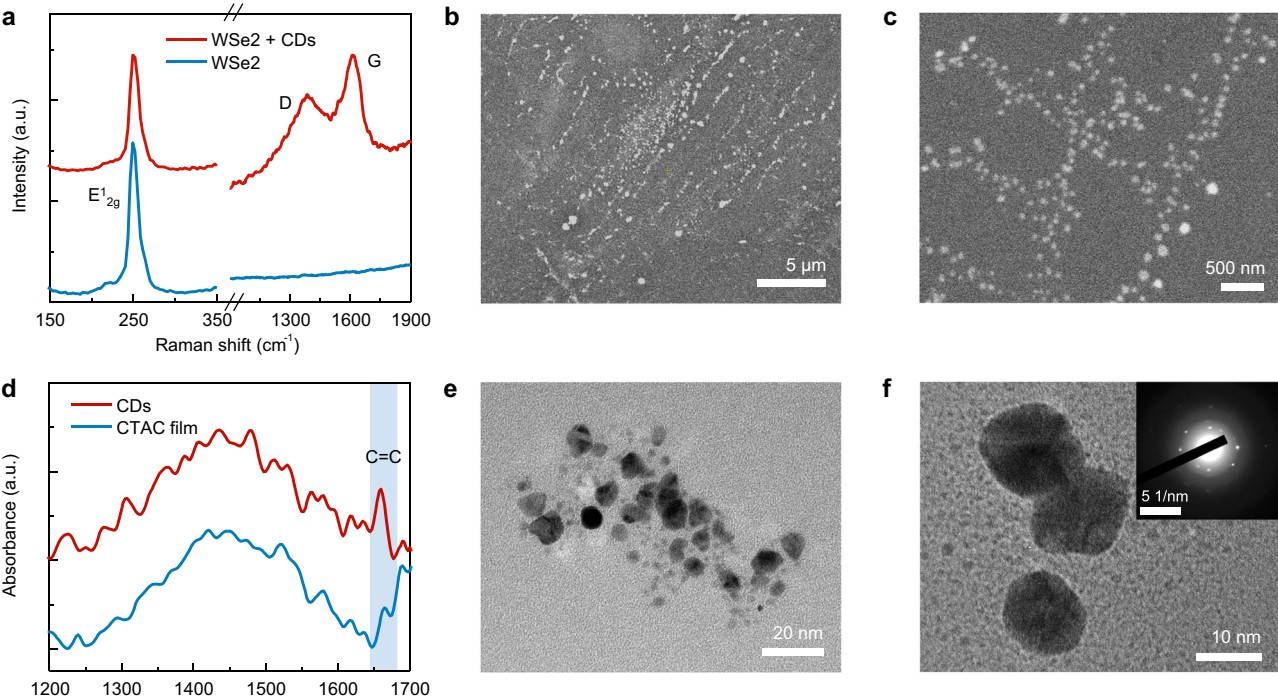

**Fig. 3 | Material characterizations of optically synthesized CDs. a** Raman spectra of WSe$_2$ and WSe$_2$ + CDs hybrids. **b**, **c** SEM images of the synthesized CDs. **d** Near-field nano-FTIR spectra of the CDs and pristine CTAC films. The light blue shading indicates the C=C bond spectrum regime. **e**, **f** High-resolution TEM images of the synthesized CDs. Inset in **f** shows the selected area electron diffraction (SAED) pattern of the CDs. "a.u." in (**a**, **d**) stands for arbitrary units.

CDs formation and materials characterizations are presented in Fig. 3. The optically generated CDs show pronounced broadband PL emission centered at ~ 600 nm under the excitation of a 532 nm laser (Fig. 2b, red curve). Additionally, the PL peak from WSe$_2$ exhibits a clear redshift from ~750 nm to ~780 nm, resulting from the charge transfer between the CDs and WSe$_2$[22,23]. Due to the minimal light absorption of CTAC and monolayer WSe$_2$ at the laser wavelength and negligible laser-induced temperature rise (Supplementary Fig. 1), we preclude the contribution of photothermal effects. Instead, this light-driven reaction is ascribed to the WSe$_2$-catalyzed C–H activation, which will be discussed later.

The photochemical reaction rate for the synthesis of CDs can be tuned by two orders of magnitude by controlling the laser power (Fig. 2c and Supplementary Movie 1). We also demonstrated the CD synthesis with a larger laser spot (Supplementary Fig. 2). Under low-power laser irradiation, the emission of synthesized CDs remains stable for more than 20 min (Supplementary Fig. 3). Besides WSe$_2$, we also demonstrate the light-driven C–H activation and generation of CDs from CTAC on CVD-grown WS$_2$ and MoS$_2$ monolayers (Fig. 2d, e). Similar orangish PL emission from CDs can be directly visualized in optical imaging (Inset in Fig. 2d). The PL spectra of MoS$_2$/WS$_2$ + CDs hybrids also showed similar features, including a broadband emission from CDs centered at ~600 nm and a redshifted peak from MoS$_2$/WS$_2$. In addition, under the 660 nm laser excitation, the PL spectra from the WSe$_2$/WS$_2$ + CDs hybrids are distinct from those under the 532 nm excitation (Fig. 2f). This excitation wavelength-dependent PL emission is a characteristic feature of CDs[24,25].

The light-driven, 2D TMDC-mediated synthesis of CDs is confirmed by multiple characterization techniques. The Raman spectrum shows a *D* band at ~1380 cm$^{-1}$ and a *G* band at ~1600 cm$^{-1}$ (Fig. 3a), which are signatures of CDs[26]. The scanning electron microscope (SEM) images also reveal the existence of CD nanoparticles in the laser-irradiated areas (Fig. 3b, c). The as-synthesized CDs have a size distribution of 5–15 nm, as shown in the transmission electron

microscope (TEM) images (Fig. 3e, f). Such large size distribution is consistent with the broad PL emission bands (Fig. 2b), which may mask the size-dependent PL properties from the carbon core[27,28]. The selected-area electron diffraction pattern exhibits bright diffraction spots and amorphous rings (inset in Fig. 3f), indicating a semi-crystalline structure of CDs. The chemical composition of CDs is further examined by a near-field nanoscale Fourier transform infrared spectroscopy (nano-FTIR). Compared to the pristine CTAC film, the nano-FTIR spectrum of CDs presents a prominent absorption band at ~1660 cm$^{-1}$ (Fig. 3d), which is assigned to the vibrations of C=C bonds in CDs[29].

Next, we discuss the underlying mechanisms of the light-driven C–H bond activation medicated by 2D TMDCs. C–H activation requires a sufficiently negative hydrogen adsorption-free energy[30]; however, pristine 2D TMDCs usually cannot meet this prerequisite since they are known to be facile hydrogen evolution materials[31]. To identify the potential active sites in our study that drive the C–H bond activation, we first measured the X-ray photoelectron spectroscopy spectra of the monolayer WSe$_2$. The results indicate the existence of prevalent Se vacancies and O adsorption on the CVD-grown WSe$_2$ surfaces (Supplementary Fig. 4)[32,33]. To analyze the role of Se vacancies and O substitution on WSe$_2$, we calculated the projected density of states (PDOS) of local W-sites using DFT calculations (Fig. 4a and Supplementary Fig. 5). With the increasing number of Se vacancies, there is an obvious shift of the peak toward the Fermi level (Fig. 4b). The calculated average energies of the *d*-electrons (i.e., the *d*-band center) of the sites with Se vacancies are also closer to the Fermi level compared to a pristine WSe$_2$. According to the *d*-band center theory[34], a surface site with a *d*-band center closer to the Fermi level corresponds to a significantly stronger H adsorption capacity[35], which facilitates the C–H bond activation due to the stronger driving force to "pull" an H down to the surface[36]. Similar conclusions can be found on a WSe$_2$ surface with oxygen substitution at Se sites (Fig. 4c). Meanwhile, the existence of adsorbed oxygen and the subsequently formed hydroxyl can act as

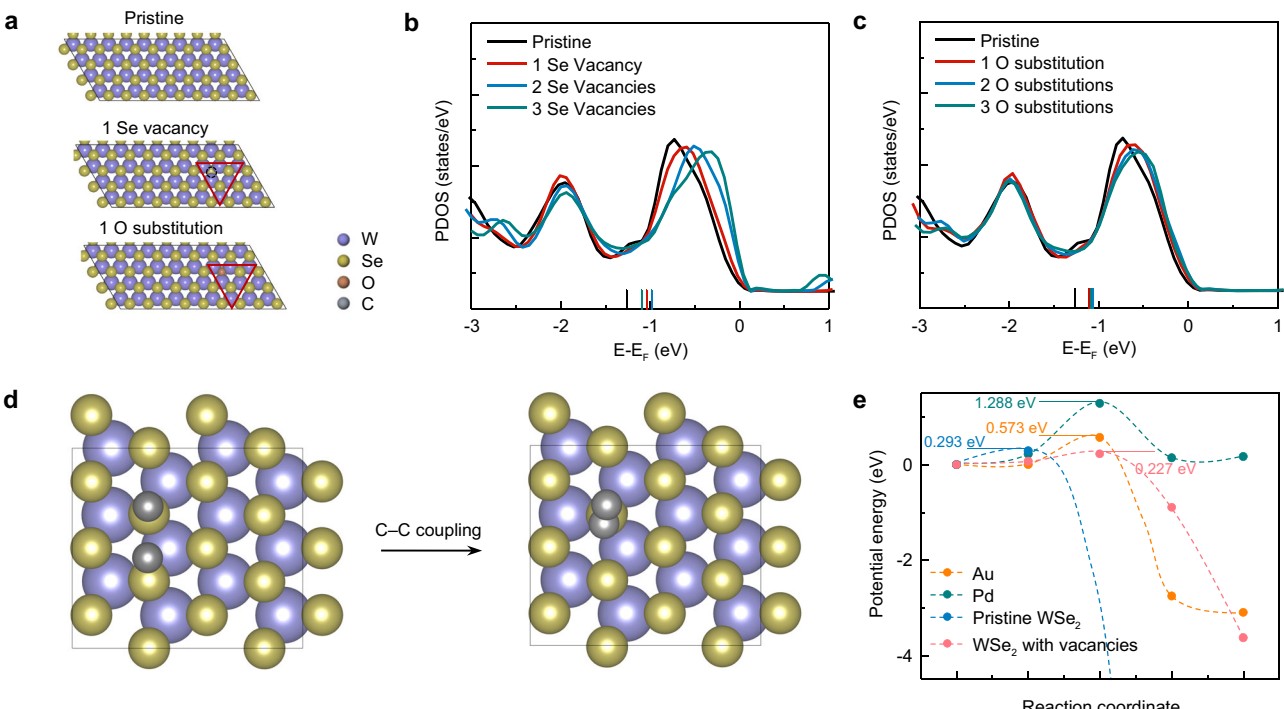

**Fig. 4 | First-principles calculations to provide insights into the light-driven C−H activation mediated by 2D materials. a** Optimized structures considered for DFT calculations. Pristine WSe₂ and WSe₂₋ₓ with Se vacancies or O substitutions are considered. **b**, **c** PDOS of the *d*-electrons of local W-sites (red triangles in a) at pristine WSe₂ and WSe₂₋ₓ with Se vacancies (**b**) or O substitutions (**c**). The vertical bars indicate the calculated *d*-band centers. The Fermi levels are shifted to zero. **d** The process of C−C coupling considered for DFT calculations on the WSe₂ surface. **e** Comparison of the kinetic barriers of C−C coupling on the WSe₂ and other surfaces.

the promoters to expedite C−H activation due to a facile O/HO-promoted mechanism[37–40]. To verify the theoretical hypothesis, we conducted control experiments on mechanically exfoliated WSe₂ flakes with fewer surface defects[41], and the results show that a much higher optical power is required for this reaction to occur (Supplementary Fig. 6). We also performed control experiments on graphene without Se vacancies, where the reaction did not occur even under high optical power (Supplementary Fig. 7). These theoretical analyses and experiments indicate that the Se vacancy and O substitution in WSe₂ can both lead to a more facile C−H activation capacity due to either higher reactivity of a defected surface or an O-promotion effect.

For long carbon chains, the C−H activation is followed by the formation of C=C bonds[42]. We further investigate the capability of 2D TMDCs to drive the C=C formation. We analyze the C−C coupling on material surfaces (Fig. 4d), where two carbon atoms are bonded together. We compare the calculated kinetic energy barriers of this process for WSe₂ and other common catalyst surfaces for C−H activation (Supplementary Fig. 8), including gold (Au) and palladium (Pd). The energy barrier of C−C coupling on WSe₂ surfaces is calculated to be 0.29 eV (Fig. 4e), which is significantly lower than that on Au (0.57 eV) and Pd (1.29 eV). These results indicate that while metal catalysts (e.g., Pd and Au) are suitable for C−H activation in short-chain molecules, they cannot be generalized to long carbon chains due to the high activation energy of C−C coupling to form C=C bonds. This energy barrier is further reduced to 0.23 eV on WSe₂ surfaces with Se vacancies (Fig. 4e and Supplementary Fig. 9). Our results demonstrate the potential of 2D TMDCs as promising catalysts to drive the C−H activation of long-chain molecules and facilitate the subsequent C=C formation.

## Discussion

In summary, we discover the 2D-TMDC-mediated C−H activation in long-chain organic molecules under light illumination. Our experimental characterizations coupled with theoretical calculations reveal the role of defects and oxidized states on TMDCs in the promotion of H adsorption and C−H activation reactions. Moreover, we find that the energy barrier of C−C coupling mediated by 2D TMDCs is much lower than the commonly used metal catalysts for C−H activation of short-chain alkanes, highlighting its promising performance of C−H activation for complex molecules.

This light-controlled site-specific C−H activation also enables the optical printing of luminescent CDs on solid substrates and provides an approach toward data encryption and information technology[43]. By controlling the thickness of CTAC layer, laser power, and irradiation time, we can write CDs by laser scanning without changing the morphology of the film (Supplementary Fig. 10). Thus, the embedded patterns remain hidden under white light illumination and can be read out by fluorescence, Raman, or PL imaging (Supplementary Fig. 11). In addition, we can easily erase the synthesized CDs by rinsing the sample with water and coating the 2D TMDC with a new CTAC layer for optical rewriting of CDs (Supplementary Fig. 12). Another promising application of such light-driven synthesis of luminescent CDs is solid-state light-emitting device[44]. Besides CTAC, the 2D-TMDC-mediated light-driven C−H activation is applicable to other long-chain molecules, including polyethylene (Supplementary Fig. 13), octyltrimethylammonium chloride, and polyvinyl alcohol (Supplementary Fig. 14). We anticipate that the 2D-TMDC-mediated light-driven C−H activation in complex organic molecules will open up new possibilities for applications in chemical synthesis, photonics, the degradation of organic pollutants, and plastic recycling.

## Methods
### Chemicals and materials
CTAC was purchased from Chem-Impex. Other chemicals, including octyltrimethylammonium chloride, polyethylene, and polyvinyl

alcohol, were purchased from Sigma-Aldrich. All the materials were used without further purification. Monolayer $WSe_2$ was synthesized using tungsten hexacarbonyl $(W(CO)_6)$ and hydrogen selenide $(H_2Se)$ in a cold-wall vertical reactor with an inductively heated SiC-coated graphite susceptor. Ultrahigh purity hydrogen was used as the carrier gas, and c-plane (001) double-side polished sapphire was used as substrates[45]. Monolayer $WS_2$ was prepared by the atmospheric pressure CVD method using a tube furnace with argon as the carrier gas. Two cleaned $SiO_2/Si$ wafers sandwiched with ~10 mg $WO_3$ powders were placed in a 2 cm diameter quartz tube, which was heated up to 700 °C and held for 15 min in the furnace. Simultaneously, sulfur powders were separately heated up to 250 °C with a heating belt[46]. Monolayer $MoS_2$ was grown by CVD using a Thermo Scientific Lindberg/Blue M Tube Furnace. $MoO_3$ powder (15 mg) and sulfur powder (1 g) were loaded in a quartz tube and heated independently. After four purging cycles, the tube was filled with ultrahigh purity $N_2$ to 760 Torr. The furnace was heated to 850 °C at a rate of 50 °C $min^{-1}$ for 5-min growth and then cooled down to room temperature[47]. CVD-grown monolayer graphene was purchased from SixCarbon.

## Optical setup

The light-driven C–H activation and laser writing of CDs were performed in a Nikon inverted microscope (Nikon TiE) equipped with a ×100 oil objective (Nikon, NA 0.5–1.3), a halogen white light source (12 V, 100 W), a bright-field or dark-field condenser (NA 1.20–1.43), and a color charge-coupled device camera (Nikon). A continuous-wave 532 nm laser (Coherent, Genesis MX STM-1 W) or a continuous-wave 660 nm laser (Laser Quantum) was expanded with a 5× beam expander (Thorlabs, GBE05-A) and directed to the microscope.

## Characterizations

The Raman spectra and mapping were measured on a Renishaw system using a 532 nm wavelength laser source. The absorption spectra and PL spectra were recorded with a spectrograph (Andor) and an EMCCD (Andor) integrated into an inverted optical microscope. The scanning electron microscopy (SEM) images were taken with a FEI Quanta 650 SEM. TEM images and diffraction patterns were obtained with a JEOL 1400 (120 kV) with Gatan Inc. One view camera and a specialized TEM holder (Laser Prismatics). Near-field nano-FTIR measurements were performed with a commercial Neaspec system equipped with a broadband laser source[48]. The XPS spectra were collected on a Kratos AXIS Ultra XPS spectrometer.

## DFT calculations

All DFT calculations were performed using the VASP code with the valence electrons treated by expanding the Kohn-Sham wave functions in a plane-wave basis set[49]. The method of generalized gradient approximation using the revised Perdew–Burke–Ernzerhof functional was employed to describe the electronic exchange and correlations[50]. The core electrons were treated by the projector augmented wave method[51]. Van der Waals corrections were included within Grimme's framework (DFT + D3)[52]. Convergence was defined when the forces of each atom fell below 0.05 eV per Å. The energy cutoff was set to 400 eV. A $(3 \times 3 \times 1)$ k point mesh was employed to sample the Brillouin zone based on the method of Monkhorst and Pack[53]. The kinetic barriers were calculated based on the climbing-image nudged elastic band method[54]. To ensure sufficient spacing, we placed a vacuum spacing of at least 12 Å perpendicular to the surface.

## Data availability

All data that support the findings of this study are included in the paper and/or Supplementary Information.

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

## Acknowledgements

Y.Z. acknowledges the financial support of the National Institute of General Medical Sciences of the National Institutes of Health (DP2GM128446) and the National Science Foundation (NSF-ECCS-2001650). H.L. and D.Z. acknowledges the JSPS KAKENHI (grant no. JP23K13703), the Hirose Foundation, and the AIMR Fusion Research. J.L. acknowledges the financial support of the University Graduate Continuing Fellowship from The University of Texas at Austin. D.Z. at Tohoku University also acknowledges the support of Shanghai Jiao Tong University Outstanding Doctoral Student Development Fund and National Natural Science Foundation of China (no. 22309109). J.M.R and H.Z. acknowledge the financial support of the National Science Foundation through the Penn State 2D Crystal Consortium—Materials Innovation Platform (2DCC-MIP) under NSF cooperative agreement DMR-2039351. J.M.L. acknowledges Baylor University for financial support through startup funds. B.B acknowledges support from the NSF Graduate Research Fellowship (DGE 2146752). TEM imaging work was funded by the Soft Matter Electron Microscopy Program (KC11BN), supported by the Office of Science, Office of Basic Energy Science, US Department of Energy, under Contract DE-AC02-05CH11231. The authors acknowledge the Center for Computational Materials Science, Institute for Materials Research, Tohoku University for the use of MASAMUNE-IMR (project no. 202312-SCKXX-0203 and 202312-SCKXX-0207) and the Institute for Solid State Physics (ISSP) at the University of Tokyo for the use of their supercomputers.

## Author contributions

J.L., Y.Z., and H.L. conceived the idea and designed the research. J.L. prepared the samples, worked on the experiments, and collected the data. H.L., D.Z., and Z.G. worked on the DFT calculations. Z.C. assisted in the revision. X.J., B.B., M.C., A.M.M., and C.P.G. worked on TEM imaging. J.M.L. and R.K. performed the nano-FTIR measurement. H.Z. and J.M.R. synthesized WSe₂. T.Z. and M.T. synthesized WS₂. Y.G. synthesized MoS₂ under the supervision of D.A. Z.W. and S.H. assisted in sample preparation and experiments. Y.Z. supervised the project. J.L., H.L., and Y.Z. wrote the manuscript with input from all authors.

## Competing interests

The authors declare no competing interests.
