## [Peer Review File · Nature Communications]

Light-driven C-H activation mediated by 2D transition metal dichalcogenidesReviewer #1 (Remarks to the Author):

I appreciate the efforts from the authors in addressing my questions. The manuscript has been approved after the revision. However, some problems are still not well addressed. The manuscript is not well supported. I do not recommend its publication in Nature Communications.

1. The authors claimed that "we preclude the possibility of photon-induced or photothermal effects". I do not think the photo-induced or photothermal effects could be precluded. The authors did not provide convincing evidence. The authors used 0.2mW for PL excitation, which is quite high for TMD monolayers and TMD monolayers showed quite strong light-matter interactions and quite high absorption.

2. The authors cited one ref to show that the PL of carbon dots is not necessarily quantum size dependent. But some previous refs also showed that some quantum dots really showed size-dependent PL emission (eg. DOI:10.1039/C2TC00140C). The author needs to compare their carbon dots with previously reported ones and show people the key difference or similarity. In particular, the authors need to well explain the reasons why their carbon dots would not show size dependent PL.

3. I do believe that other types of 2D materials without S- or Se- vacancies, should be used as a control.

Reviewer #2 (Remarks to the Author):

The revision looks good. The authors have addressed the questions. I would like to support it for the publication in Nature Communications.

Reviewer #3 (Remarks to the Author):

In this manuscript, the authors reported the light-driven C-H activation in complex organic materials mediated by 2D transition metal dichalcogenides (TMDCs) and the resultant solid-state synthesis of luminescent carbon dots in a spatially-resolved fashion. The experimental characterizations coupled with theoretical calculations reveal the role of defects and oxidized states on TMDCs in the promotion of H adsorption and C-H activation reactions. This light-controlled site-specific C-H activation also enables optical printing of luminescent carbon dots on solid substrates and provides an approach towards data encryption and information technology. This work is interesting, but there are still some issues that are unclear, many of them are needed to be discussed in depth.

The reviewer recommends its publication in Nature communications but major revisions as listed below are required before acceptance.

1. Did the authors use a CW laser or a pulsed laser for the experiment? If so, how could you prove that the reaction was not caused by thermal efficiency? The authors should note that laser radiation will generate heat, and heating is also one of the methods for the preparation of carbon dots. The authors need to give convincing evidence to exclude the possibility of thermally induced carbon dots.

2. In Figure 2, under the excitation of 532 nm, the emission spectra of carbon dots generated by different metal compounds (WSe₂, WS₂, MoS₂) are very different, and the spectrum of WSe₂ also has a strong emission at 770 nm, please explain these two phenomena.

3. What is the laser spot area currently used by the authors? Have the authors tried the synthesis of carbon dots under a larger spot.

4. Since the authors mentioned that WSe₂ catalyze C-H bond activation, I would like to know if the

chemical properties of WSe₂ remain unchanged after laser and how many times can the author perform the erasing and rewriting experiment.

5. Why was the C-H activation reaction carried out with cetyltrimethylammonium chloride instead of pure alkane?

Reviewer #1 (Remarks to the Author):

I appreciate the efforts from the authors in addressing my questions. The manuscript has been approved after the revision. However, some problems are still not well addressed. The manuscript is not well supported. I do not recommend its publication in Nature Communications.

Response: We appreciate the reviewer's positive comments and thank you for pointing out the existing concerns to help us further improve our work. We have revised our manuscript based on the comments to address these concerns, as detailed in the following point-by-point responses.

1. The authors claimed that "we preclude the possibility of photon-induced or photothermal effects". I do not think the photo-induced or photothermal effects could be precluded. The authors did not provide convincing evidence. The authors used 0.2mW for PL excitation, which is quite high for TMD monolayers and TMD monolayers showed quite strong light-matter interactions and quite high absorption.

Response: Thank you for the comment. Although a 0.2 mW focused laser might be high for TMD materials, our laser is far away from their exciton wavelengths. Our measurement shows less than 2 % optical absorption at the laser wavelengths we used (Fig. R1a). With such minimal light absorption, laser heating effects can be negligible. To support our claim, we further used COMSOL to simulate the temperature increase under laser irradiation. For a 532 nm laser with 0.2 mW power and 0.8 μm beam diameter (Gaussian intensity profile), the temperature increase is less than 10 K (Fig. R1c), which is negligible to drive the reaction. Our new control experiments in Fig. R2 also suggest the thermal effect can be neglected, as detailed in the response to comment #3. In the revised manuscript, we added the temperature simulation results in Supplementary Materials and revised the claim to be clearer.

Changes made:

In Supplementary Materials: Supplementary Fig. 1 was updated as below, with new simulation results and discussion.

Page 4, Lines 82-84: "Due to the negligible light absorption of CTAC and monolayer WSe₂ (Supplementary Fig. 1), we preclude the contribution of photothermal effects." was replaced by "Due to the minimal light absorption of CTAC and monolayer WSe₂ at the laser wavelength and negligible laser-induced temperature rise (Supplementary Fig. 1), we preclude the contribution of photothermal effects."

Fig. R1 (also see Supplementary Fig. 1). **a**, Measured optical absorption of CTAC and monolayer WSe₂. The green and red vertical lines indicate the laser wavelengths used in this work, 532 nm and 660 nm, respectively. **b**, Schematic of the simulation setup. **c**, COMSOL simulation of (left) 3D and (right) top-view temperature distribution under 532 nm laser heating. The laser power is 0.2 mW with a beam diameter of 800 nm. Due to the very low optical absorption at the laser wavelength (< 2%), the local temperature increase induced by the laser is less than 10 K.

2. The authors cited one ref to show that the PL of carbon dots is not necessarily quantum size dependent. But some previous refs also showed that some quantum dots really showed size-dependent PL emission (eg. DOI:10.1039/C2TC00140C). The author needs to compare their carbon dots with previously reported ones and show people the key difference or similarity. In particular, the authors need to well explain the reasons why their carbon dots would not show size dependent PL.

Response: Thanks for the comments. Carbon dots have been found to have both size-dependent and size-independent PL properties, which depend on the type of carbon dots and synthesis methods. Size-dependent PL usually comes from the carbon core, which is more commonly observed in carbon dots with a narrow size distribution, while size-independent PL is mainly governed by the surface states with a large size distribution and broad emission bands. It is generally accepted that carbon dots are different from

semiconductor quantum dots, and their PL usually shows size independence and excitation dependence (*J. Phys. Chem. Lett.* 2019, *10*, 5182-5188; *Nano Res.* 2015, *8*, 355-381). For the reference the reviewer mentioned (DOI:10.1039/C2TC00140C, *J. Mater. Chem. C* 2013, *1*, 580-586), narrowly-dispersed carbon dots were prepared by a unique approach from single polymeric nanoparticles. In addition, they show a different trend than conventional semiconductor quantum dots (smaller carbon dots have red-shifting PL), which is contrary to the quantum confinement effect. The authors also discussed that the size-dependency is complicated and not general for all carbon dots.

In our case, we prepare carbon dots via laser-induced reactions from solid surfactant precursors. From the SEM/TEM images (Fig. 3c,e) and PL spectra (Fig. 2b,d-f), our carbon dots show large size distribution and broad PL emission bands, which may mask the size-dependent PL properties. In addition, the broad PL emission from carbon dots has been observed from similar preparation methods, such as laser synthesis (*Adv. Mater.* 2019, *31*, 1901371) and direct carbonization (*Nanoscale* 2018, *10*, 21492-21498). In the revised manuscript, we added relevant discussion to avoid any confusion on the size-dependent PL.

Changes made:

Pages 5-6, Lines 111-113: “Such large size distribution is consistent with the broad PL emission bands (Fig. 2b), which may mask the size-dependent PL properties from the carbon core^{27,28}.” **was added.**

27. Zhu, S. et al. The photoluminescence mechanism in carbon dots (graphene quantum dots, carbon nanodots, and polymer dots): current state and future perspective. *Nano Res.* **8**, 355-381 (2015).

28. Tao, S., Feng, T., Zheng, C., Zhu, S. & Yang, B. Carbonized Polymer Dots: A Brand New Perspective to Recognize Luminescent Carbon-Based Nanomaterials. *J. Phys. Chem. Lett.* **10**, 5182-5188 (2019).”

3. I do believe that other types of 2D materials without S- or Se- vacancies, should be used as a control.

Response: Thanks for the suggestion. To illustrate the role of S- or Se- vacancies, we added a control experiment with another 2D material, graphene, which has no S- or Se- vacancies. As shown in Fig. R2 below (also see Supplementary Fig. 7 of the revised manuscript), we irradiated a high-power laser (~ 10 mW) on the CTAC + graphene sample. Despite the local heating that caused the melting of the CTAC layer (Fig. R2b), no carbon dots were formed, and no PL emission was observed (Fig. R2c). This result also supports that the thermal effect is not important in this light-driven reaction. We added this control experiment in the revised manuscript.

Changes made:

Page 7, Lines 144-146: “We also performed control experiments on graphene without Se vacancies, where the reaction did not occur even under high optical power (Supplementary Fig. 7).” was added.

Page 10, Line 195: “CVD-grown monolayer graphene was purchased from SixCarbon.” was added.

In Supplementary Materials: Supplementary Fig. 7 was added as below.

Fig. R2 (also see Supplementary Fig. 7). Control experiments on CVD-grown graphene. a, Optical image showing the graphene coated with a thin layer of CTAC before laser irradiation. **b,** After a 10-mW laser irradiation for seconds, the laser heating caused the local melting of the CTAC layer, as indicated in the white circle. The laser position is at the crosshair. **c,** No obvious PL emission from carbon dots in the dark-field image.

Reviewer #2 (Remarks to the Author):

The revision looks good. The authors have addressed the questions. I would like to support it for the publication in Nature Communications.

Response: We appreciate the reviewer's positive comments and thank you for supporting the publication of our manuscript!

Reviewer #3 (Remarks to the Author):

In this manuscript, the authors reported the light-driven C-H activation in complex organic materials mediated by 2D transition metal dichalcogenides (TMDCs) and the resultant solid-state synthesis of luminescent carbon dots in a spatially-resolved fashion. The experimental characterizations coupled with theoretical calculations reveal the role of defects and oxidized states on TMDCs in the promotion of H adsorption and C-H activation reactions. This light-controlled site-specific C-H activation also enables optical printing of luminescent carbon dots on solid substrates and provides an approach towards data encryption and information technology. This work is interesting, but there are still some issues that are unclear, many of them are needed to be discussed in depth.

The reviewer recommends its publication in Nature communications but major revisions as listed below are required before acceptance.

Response: We appreciate the reviewer's positive comments and thank you for supporting the publication of our manuscript! We have revised our manuscript further based on the comments below to address the concerns, as detailed in the following point-by-point responses.

1. Did the authors use a CW laser or a pulsed laser for the experiment? If so, how could you prove that the reaction was not caused by thermal efficiency? The authors should note that laser radiation will generate heat, and heating is also one of the methods for the preparation of carbon dots. The authors need to give convincing evidence to exclude the possibility of thermally induced carbon dots.

Response: Thank you for the comment. This comment is similar to the comment #1 from Reviewer #1. We used a continuous-wave (CW) laser for the experiments. The lowest optical power to drive the reaction is 0.2 mW. Due to low optical absorption at less than 2 %, laser heating effects can be neglected. To support this claim, we further used COMSOL to simulate the temperature increase under laser irradiation. For a 532 nm laser with 0.2 mW power and 0.8 μm beam diameter (Gaussian intensity profile), the temperature increase is less than 10 K (Fig. R1c), which is negligible to drive the reaction. Our new control experiments in Fig. R2 also suggest the thermal effect can be neglected. In the revised manuscript, we added the temperature simulation results in Supplementary Materials and revised the claim to be clearer.

Changes made:

In Supplementary Materials: Supplementary Fig. 1 was updated as below, with new simulation results and discussion.

Page 4, Lines 82-84: “Due to the negligible light absorption of CTAC and monolayer WSe₂ (Supplementary Fig. 1), we preclude the contribution of photothermal effects.” was replaced by “Due to the minimal light absorption of CTAC and monolayer WSe₂ at the laser wavelength and negligible laser-induced temperature rise (Supplementary Fig. 1), we preclude the contribution of photothermal effects.”

Page 10, Line 199: “continuous-wave” was added.

Fig. R1 (also see Supplementary Fig. 1). **a**, Measured optical absorption of CTAC and monolayer WSe₂. The green and red vertical lines indicate the laser wavelengths used in this work, 532 nm and 660 nm, respectively. **b**, Schematic of the simulation setup. **c**, COMSOL simulation of (left) 3D and (right) top-view temperature distribution under 532 nm laser heating. The laser power is 0.2 mW with a beam diameter of 800 nm. Due to the very low optical absorption at the laser wavelength (< 2%), the local temperature increase induced by the laser is less than 10 K.

2. In Figure 2, under the excitation of 532 nm, the emission spectra of carbon dots generated by different metal compounds (WSe₂, WS₂, MoS₂) are very different, and the spectrum of WSe₂ also has a strong emission at 770 nm, please explain these two phenomena.

Response: Thank you for the comment. Under 532 nm laser excitation, the optically generated carbon dots all show broadband PL emission centered at ~ 600 nm. The PL spectra in Fig. 2b,d,e (also plotted below for better visualization) contain both the carbon dots emission (broadband) and the emission from monolayer TMDC (a narrower peak). The emission at ~770 nm from the WSe₂ + CDs hybrids comes from the monolayer WSe₂ (Fig. 2b). Similarly, the emission peak at ~ 670 nm from the WS₂ + CDs hybrids comes from the monolayer WS₂ (Fig. 2d). The peak from monolayer MoS₂ at ~ 690 nm is less obvious and masked by the carbon dots emission band (Fig. 2e). The PL peak from monolayer TMDCs in the carbon dots/TMDC hybrids is redshifted compared to the PL peak from the bare TMDCs, which is due to the charge transfer between carbon dots and monolayer TMDCs (*Adv. Mater.* 2019, 31, 1903613; *J. Phys. Chem. C* 2017, 121, 1997-2004). We have included these discussions in the manuscript to explain these phenomena.

Page 4, Lines 79-81: “The optically generated CDs show pronounced broadband PL emission centered at ~ 600 nm under the excitation of a 532 nm laser (Fig. 2b, red curve). Additionally, the PL peak from WSe₂ exhibits a clear redshift from ~750 nm to ~780 nm, resulting from the charge transfer between the CDs and WSe₂^{22,23}.”

Page 4, Lines 92-93: “The PL spectra of MoS₂/WS₂ + CDs also showed similar features, including a broadband emission from CDs centered at ~ 600 nm and a redshifted peak from MoS₂/WS₂.”

Fig. R3 (part of Fig. 2 in manuscript). The PL spectra of (b) WSe₂ and WSe₂ + CDs hybrids; (d) WS₂ and WS₂ + CDs hybrids; and (e) MoS₂ and MoS₂ + CDs hybrids under the excitation of a 532 nm laser.

3. What is the laser spot area currently used by the authors? Have the authors tried the synthesis of carbon dots under a larger spot.

Response: Thank you for the suggestion. We used a $\times 100$ oil objective with a numerical aperture (NA) of 0.5-1.3 (see Methods). The smallest laser beam size is $\sim 0.8 \mu\text{m}$ (see Fig. 2a). We also tried the synthesis of carbon dots with $\times 40$ and $\times 20$ objectives, with a larger laser spot up to $\sim 5 \mu\text{m}$ (Fig. R4, also see Supplementary Fig. 2 of the revised manuscript). We added this experiment to the revised manuscript.

Changes made:

Page 4, Lines 87-88: “We also demonstrated the CD synthesis with a larger laser spot (Supplementary Fig. 2).” was added.

In Supplementary Materials: Supplementary Fig. 2 was added as below.

Fig. R4 (also see Supplementary Fig. 2). Light-driven synthesis of CDs with larger laser spots. **a**, $\times 100$ objective; **b**, $\times 40$ objective; **c**, $\times 20$ objective. All scale bars are $5 \mu\text{m}$.

4. Since the authors mentioned that WSe₂ catalyze C-H bond activation, I would like to know if the chemical properties of WSe₂ remain unchanged after laser and how many times can the author perform the erasing and rewriting experiment.

Response: Thanks for the comment. The WSe₂ after CD synthesis remains intact. The erasing and rewriting can be repeated many times by simply rinsing the sample and coating another layer of precursor materials. We added a demonstration for the erasing and rewriting experiments (Fig. R5, also see Supplementary Fig. 12 of the revised manuscript), where a scratch region is introduced as the reference location. We used the computer to control the stage to synthesize CDs at the same location. A $\times 40$ objective with a relatively large laser spot ($\sim 2 \mu\text{m}$) was used. We added this discussion and experiment to the revised manuscript.

Changes made:

Page 9, Lines 181-183: “In addition, we can easily erase the synthesized CDs by rinsing the sample with water and coating the 2D TMDC with a new CTAC layer for optical rewriting of CDs (Supplementary Fig. 12).” was added.

In Supplementary Materials: Supplementary Fig. 12 was added as below.

Fig. R5 (also see Supplementary Fig. 12). Erasing and rewriting of CDs. CDs were synthesized by laser irradiation on WSe₂ + CTAC sample at the same location three times. The location was marked by the reference scratch/dot and the white arrows. Bright PL emission was clearly observed in all three experiments. Scale bar: 10 μm.

5. Why was the C-H activation reaction carried out with cetyltrimethylammonium chloride instead of pure alkane?

Response: Thank you for the suggestion. We worked on cetyltrimethylammonium chloride (CTAC) as the first example due to its clean carbon-chain structure, solid form under ambient conditions, and wide existence in nanomaterials systems. We also demonstrated the 2D TMDC-mediated reaction as a general strategy for other long-carbon-chain molecules, such as polyvinyl alcohol. Following the reviewer's suggestion, we also explored the possibility of light-driven C-H activation and CD synthesis using pure alkane and polyethylene (Fig. R6, also see Supplementary Fig. 13 of the revised manuscript), where similar PL emission was observed. We added this experiment to the revised manuscript.

Changes made:

Page 9, Lines 183-184: “the 2D-TMDC-mediated light-driven C-H activation is applicable to other long-chain molecules, such as polyethylene (Supplementary Fig. 13).” was added.

In Supplementary Materials: Supplementary Fig. 13 was added as below.

Fig. R6 (also see Supplementary Fig. 13). Light-driven C-H activation and CD synthesis with polyethylene + WSe₂. **a**, Optical imaging of a polyethylene film on a monolayer WSe₂. **b**, CD synthesis under laser irradiation. Laser power is 3.2 mW. **c**, Measured PL spectra from polyethylene (blue) and polyethylene + WSe₂ sample (red). The latter shows obvious broadband emission band from CDs. Scale bar: 5 μ m.

Reviewer #1 (Remarks to the Author):

The authors have well address my concerns. Thanks. I would recommend its publication.

Reviewer #3 (Remarks to the Author):

The quality of the revised manuscript has been greatly improved and this manuscript is acceptable with minor revision of adding some recently closely related references (Nat Commun 15, 3043 (2024); Appl. Phys. Rev. 11, 011417 (2024); Angewandte Chemie 135, e202301651(2023)).

Reviewer #1 (Remarks to the Author):

The authors have well address my concerns. Thanks. I would recommend its publication.

Response: We appreciate the reviewer's positive feedback and constructive comments, and thank you for supporting the publication of our manuscript.

Reviewer #3 (Remarks to the Author):

The quality of the revised manuscript has been greatly improved and this manuscript is acceptable with minor revision of adding some recently closely related references (Nat Commun 15, 3043 (2024); Appl. Phys. Rev. 11, 011417 (2024); Angewandte Chemie 135, e202301651(2023)).

Response: We appreciate the reviewer's positive feedback and constructive comments, and thank you for supporting the publication of our manuscript. In the revised manuscript, we added a discussion on the potential applications of light-driven synthesis of carbon dots for photonic LED devices with suggested reference properly cited.